# Discrepancy Ratio: Evaluating Model Performance When Even Experts Disagree on the Truth

**Igor Lovchinsky**      **Alon Daks**      **Israel Malkin**      **Pouya Samangouei**      **Ardavan Saeedi**
ilovchinsky      adaks      imalkin      psamangouei      asaeedi

**Yang Liu**      **Swami Sankaranarayanan**      **Tomer Gafner**      **Benjamin Sternlieb**
yliu      ssankaranarayanan      tgafner      bsternlieb

**Patrick Maher**                          **Nathan Silberman**
pmaher                                    nsilberman

**Butterfly Network**
@butterflynetwork.com

## Abstract

In most machine learning tasks unambiguous ground truth labels can easily be acquired. However, this luxury is often not afforded to many high-stakes, real-world scenarios such as medical image interpretation, where even expert human annotators typically exhibit very high levels of disagreement with one another. While prior works have focused on overcoming noisy labels during training, the question of how to *evaluate* models when annotators disagree about ground truth has remained largely unexplored. To address this, we propose the *discrepancy ratio*: a novel, task-independent and principled framework for validating machine learning models in the presence of high label noise. Conceptually, our approach evaluates a model by comparing its predictions to those of human annotators, taking into account the degree to which annotators disagree with one another. While our approach is entirely general, we show that in the special case of binary classification, our proposed metric can be evaluated in terms of simple, closed-form expressions that depend only on aggregate statistics of the labels and not on any individual label. Finally, we demonstrate how this framework can be used effectively to validate machine learning models using two real-world tasks from medical imaging. The discrepancy ratio metric reveals what conventional metrics do not: that our models not only vastly exceed the average human performance, but even exceed the performance of the best human experts in our datasets.

## 1 Introduction

The canonical supervised machine learning paradigm assumes the presence of both inputs and their corresponding, unambiguous outputs. In practice, the vast majority of datasets exhibit some degree of label noise. Acknowledgement of this fact has resulted in the development of algorithms tailored to training models in the presence of noisy ground truth labels (Sukhbaatar et al., 2015; Frenay & Kaban, 2014; Zhu & Wu, 2004). These approaches include cleansing noisy labels before training begins (Hodge & Austin, 2004), making the training process robust to noise (Patrini et al., 2017; Vahdat, 2017; Rolnick et al., 2017) and actively modeling label noise (Sukhbaatar et al., 2015; Joseph et al., 1995). These methods almost exclusively address only the problem of noisy labels during training and tend to assume that at least *some* noise-free ground truth is available at test time in order to properly evaluate different approaches. Indeed, without an unambiguous yard-stick, how can competing schemes possibly be evaluated in a standardized manner?

There exist many domains, such as medical imaging (Rosenkrantz et al., 2013; Lazarus et al., 2006) and machine translation (Wang et al., 2018), in which access to unambiguous ground truth, even

at *evaluation time*, is not practically achievable. In these domains not only do annotators disagree with each other, but they often even disagree with themselves (Gulshan et al., 2016; Hansegard et al., 2009). This, however, does not imply that the annotators are not trustworthy, but rather that these tasks are difficult even for domain experts. Datasets in these regimes are, in practice, quite different from those typically used in machine learning research (which are usually geared towards content that can be easily curated by lay people). As a consequence, studies utilizing public data often resort to simulating label noise. Indeed, a theoretical understanding of evaluation in the presence of ambiguous test data has remained largely unexplored.

In this work, we propose a new framework for evaluating the performance of machine learning models in the presence of high label noise. Our main contributions are as follows:

1. We propose a novel, task-independent metric, the *discrepancy ratio*, that allows us to a) quantify model performance, taking into account how much humans disagree with each other, and b) set a precise threshold at which the model performance can be said to exceed the average human performance.

2. We perform a detailed analysis of the discrepancy ratio for the special case of binary classification and demonstrate that it can be evaluated in terms of simple, closed-form expressions that depend only on aggregate statistics (mean and variance) of the annotations and not on any individual annotation.

3. We demonstrate the effectiveness of our approach on models that we trained on two important medical imaging tasks with real-world clinical data and show that it reveals that our models not only significantly exceed the average human performance, but even beat the best human experts in our datasets.

## 2 Related Work

There are several well-known strategies for dealing with ambiguous or noisy evaluation labels. The most popular approach is to simply use a single annotator (Jafari et al., 2019; Rajchl et al., 2017; Cobzas et al., 2007; Roth et al., 2018). An approach taken by many popular, large-scale datasets, such as ImageNet (Deng et al., 2009), Places (Zhou et al., 2014), COCO (Lin et al., 2014) and Open Images V4 (Kuznetsova et al., 2018), is to take the average or majority vote. These efforts typically employ crowdsourcing in some fashion, since the labeling of this data does not require expert-level knowledge. While majority vote reduces label noise, in domains that require specialized knowledge it is often not practical to get enough experts that the noise is removed to the point where the annotations can be assumed to be "correct" (in the sense that MNIST annotations are assumed to be correct). A related approach obtains a ground-truth dataset by enforcing a resolution criterion on labels where annotators disagree. This criterion can, for example, involve a live session where annotators deliberate and come to a consensus or an adjudication process where a third party (usually a more senior, expert annotator) is tasked with resolving disagreements (Krause et al., 2018; Hannun et al., 2019; Rajpurkar et al., 2018b). This approach may be effective at removing erroneous labels (reducing variance) at the expense of annotator independence (increasing bias).

The techniques outlined above fall short in providing a unified method that can both a) be compared across different studies/tasks and b) answer the essential question of how the model performance relates to an *average* human annotator (i.e. how a prototypical human currently performs at a task). In developing high-stakes, real-world machine learning systems, the goal of evaluating a model is often to determine whether performance is safely substitutable for individual human behavior rather than comparing to an idealized oracle.

Several studies have compared model performance with inter-annotator variability. Some have visualized model performance and inter-annotator variability (e.g. via ROC curves (Krause et al., 2018; Gulshan et al., 2016) or Bland-Altman plots (Hansegard et al., 2009; Altman & Bland, 1983)) without providing an explicit metric. Several works have constructed a metric by comparing the model to the average annotator performance, though each has done so in a completely different way (Wei et al., 2019; Hannun et al., 2019; Rajpurkar et al., 2018a;b; 2017). These metrics, however, generally depend on the number of annotators used in the study and/or require that all annotators label the entire dataset (see Section 4 and Appendix C for more details). Moreover, the metrics are task-specific and cannot be interpreted outside the context of a particular study.

## 3 THE DISCREPANCY RATIO METRIC

In this section, we define the discrepancy ratio and show that it has numerous desirable properties that overcome the limitations of prior art detailed in Section 2.

### 3.1 DEFINITIONS

**Definition 1.** *For any two arbitrary sets of labels $\mathcal{Y} = \{y_1, ...y_n\}$ and $\mathcal{Y}' = \{y'_1, ...y'_{n'}\}$, the mean pairwise deviation (MPD) is defined by:*

$$\psi(\mathcal{Y}, \mathcal{Y}') = \sum_{y \in \mathcal{Y}} \sum_{y' \in \mathcal{Y}'} \frac{\delta(y, y')}{|\mathcal{Y}||\mathcal{Y}'|}, \tag{1}$$

where $\delta$ can be chosen to correspond to any measure of agreement appropriate to the task in question (e.g. squared error, absolute error, Jaccard Index (Jaccard, 1912), Cohen's kappa (Cohen, 1960)).

Consider a dataset of $N$ samples where we have a prediction $m^i$, for each sample $i$, by the model we are trying to evaluate. Further assume that each sample is labeled by at least two annotators and that each annotator may label the same sample multiple times. Formally, let $\mathcal{Y}_j^i = \{y_{jk}^i\}$ be the set of annotations provided by annotator $j$ on sample $i$ where $k$ indexes the annotation number. Finally, let $A^i$ indicate the number of annotators that have annotated sample $i$.

**Definition 2.** *We define the annotator discrepancy, which quantifies the average annotator disagreement, to be the average of the MPD over all annotator pairs and all samples in the dataset:*

$$\alpha(\{\mathcal{Y}_j^i\}) = \frac{1}{N} \sum_{i=1}^{N} \frac{1}{A^i(A^i - 1)} \sum_{\substack{j,j'=1 \\ j' \neq j}}^{A^i} \psi(\mathcal{Y}_j^i, \mathcal{Y}_{j'}^i). \tag{2}$$

Without access to unambiguous ground-truth labels, a natural way to define human performance is by comparing an annotator's labels to those provided by other humans on the same samples. Consequently, we use the annotator discrepancy as a measure of average human performance.

**Definition 3.** *Similarly, the model discrepancy is defined to be*

$$\mu(m^i, \{\mathcal{Y}_j^i\}) = \frac{1}{N} \sum_{i=1}^{N} \frac{1}{A^i} \sum_{j=1}^{A^i} \psi(m^i, \mathcal{Y}_j^i). \tag{3}$$

**Definition 4.** *We define the discrepancy ratio to be the ratio of the model and annotator discrepancies:*

$$\Delta = \frac{\mu}{\alpha}. \tag{4}$$

*Intuitively, this metric is the ratio of the average model-annotator disagreement and the annotator-annotator disagreement.* For example, a discrepancy ratio of $0.8$ would mean that the model is $20\%$ closer to the annotators than they are to each other. When $\Delta$ is above (below) $1$, the model performance can be said to be worse (better) than the average human performance. Since the measure of agreement $\delta$ is arbitrary, the discrepancy ratio can be thought of as a "meta-metric" that reports how the model performs (relative to humans) with respect to that particular measure of agreement. The dependence of the discrepancy ratio on the choice of $\delta$ is further explored in Section 4.2.2.

### 3.2 PROPERTIES

The discrepancy ratio metric, as formulated above, has several desirable properties:

**Dimensionless**: Dimensionless quantity whose meaning is invariant across different task categories (regression, classification, segmentation etc.) and whose value is interpretable and highly intuitive.

**Minimal dataset coverage requirements**: The discrepancy ratio does not require that all annotators label all samples (see Fig. 2 and Appendix C), which is often not practical in real-world situations,

| Sample Index | Model | Annotator 1 | Annotator 2 | ... | Annotator A |
|---|---|---|---|---|---|
| 1 | 0 | 0, 1 | 0, 0 | ... | 1, 0, 1 |
| 2 | 1 | – | 1, 1 | ... | 0 |
| ... | ... | ... | ... | ... | ... |
| N | 0 | 1 | 1, 1, 1 | ... | 1, 1, 1, 1, 1 |

Table 1: An Example Dataset for a Binary Classification Task.

but only that each sample is labeled by at least two annotators (two annotators are needed since it is meaningless to measure the annotator disagreement with only a single annotator).

**Insensitive to number of labels per-sample**: The discrepancy ratio is expressed in terms of averages over all model/annotation pairs and therefore does not depend on how many annotators are used (see Fig. 2), nor how many times each annotator labels each image. This property makes it convenient to compare the discrepancy ratio across studies.

Since the discrepancy ratio is formulated as a relative error that the model makes over that of the annotators, it can be naturally incorporated as a test statistic in a non-inferiority framework (Ahn et al., 2013), often used in medical research. The goal of non-inferiority testing is to show that a new method is not statistically worse than an existing standard, up to some chosen equivalence bound. For the discrepancy ratio, we can choose the equivalence bound to correspond to an acceptable fraction of the error that the model is allowed to make on top of the one that annotators already make (e.g. setting the equivalence bound to be 0.1 can be interpreted as saying that the model can make an error no more than 10% larger than the ones humans typically make). Additional properties of the discrepancy ratio, as well as comparisons to alternative formulations, are discussed in detail in Appendix C and E.

### 3.3 An Illustrative Example

In this section, we compute expressions for the model and annotator discrepancies on a generic binary classification task using the squared error as an agreement measure. This calculation will serve to build intuition for these metrics and can be used as a blueprint for evaluating them on other tasks.

Consider a dataset (see Table 1) containing $N$ samples, where each sample is given a binary score by a model and a set of $A$ annotators (an annotator may label an image more than once, or not at all). We assume all samples are labeled by at least two annotators. In the analysis below we adopt the convention that $y_{jk}^i$ denotes the $k$th label provided by the $j$th annotator on the $i$th sample, and removing an index denotes averaging over the values of that index. For example, $y_j^i$ is the average of the labels provided by the $j$th annotator on the $i$th sample and $y$ is the average annotation over all labels, annotators and samples. The order of the averages should be clear from the context.

As shown in Appendix B, given two sets of binary labels $\mathcal{Y} = \{y_1, y_2, ..., y_n\}$ and $\mathcal{Y}' = \{y_1', y_2', ..., y_{n'}'\}$ the MPD is given by the expression $y + y' - 2yy'$, where $y = \frac{1}{n} \sum_{k=1}^n y_k$. Using this result and Eq. 2, the annotator discrepancy for a single sample is then given by (Appendix B):

$$\alpha^i = 2y^i(1 - y^i) + \frac{2\sigma_{y^i}^2}{A^i - 1}. \tag{5}$$

where $\sigma_{y^i}^2$ is the variance of the labels $y^i$ and $A^i$ is the number of annotators that have labeled sample $i$. For the important special case where each annotator provided no more than a single label per image, this expression further simplifies to

$$\alpha^i = \frac{2A^i}{A^i - 1}\sigma_{y^i}^2, \tag{6}$$

where we see that annotator discrepancy is simply proportional to the variance of the labels. Similarly we can write the single-sample model discrepancy as

$$\mu^i = \frac{1}{A^i}\sum_{j=1}^{A^i} m^i + y_j^i - 2m^i y_j^i = m^i + y^i - 2m^i y^i. \tag{7}$$

Averaging this result over all samples, the model discrepancy becomes

$$\mu = (m + y - 2my) - 2\text{Cov}[m^i, y^i], \tag{8}$$

where the covariance is taken over the samples $i$ and $m$ and $y$ correspond to the model predictions and annotator labels averaged over the dataset. The expression in parentheses in Eq.8 is invariant under permutations of the model predictions $m^i$ and annotations $y^i$. It can be interpreted as the best baseline performance that can be expected when the model and annotators are simply guessing. The covariance term represents a correction to this baseline performance that takes into account the correlation of the predictions and labels (i.e. when the model and annotators provide correlated labels, the performance is improved). The expressions in Eqs. 5-7 only depend on aggregate statistics (i.e. mean and variance) of the labels and not on any particular annotation and can be used, for example, as metrics during model training.

## 4    EXPERIMENTS

We illustrate the ideas discussed in Section 3 on three datasets. First, we use MNIST (LeCun et al., 2010) with simulated models and annotators to objectively compare different strategies for evaluating model performance in the presence of noisy test data. Next, we train and evaluate models on two real-world, medical image datasets, which exhibit high degrees of expert annotator disagreement, and evaluate them using our framework.

### 4.1    MNIST DIGIT CLASSIFICATION

For our MNIST experiments, we use the standard train/test splits and original (10-class) labels. Rather than train models, we synthetically generate them by corrupting the ground truth labels. Each model is designated as **model X** where X designates the percentage of the ground truth which the model preserves. For example, **model 1.0** leaves the ground truth as is whereas **model 0.8** chooses $20\%$ of the labels randomly from amongst the other 9 classes.

We explicitly differentiate between true labels and observed labels, which are noisy and may not match the true labels. Our goal is to understand how different evaluation strategies fare in the presence of label noise, e.g. when we do not have direct access to the true labels. To this end, we evaluate several strategies: Naive Evaluation, Majority Vote, Relative F1 and our Discrepancy Ratio metric.

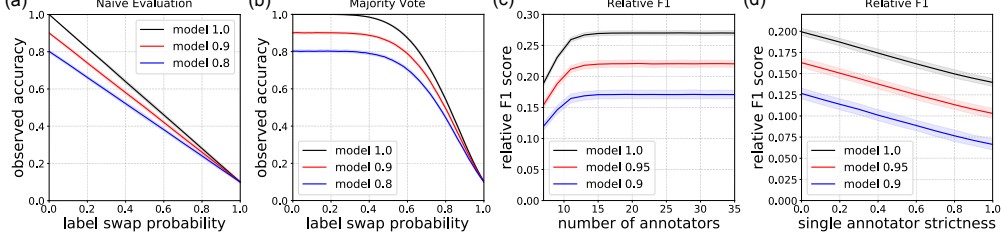

Figure 1: Performance of baseline metrics on the MNIST test set. Curves show mean value and 2-standard deviation bounds obtained from multiple simulations of the experiment. (a) Naive Evaluation exhibits linear decay of observed accuracy but cannot compare model to inter-annotator variability (b) Majority Vote is more invariant to label noise but also does not compare model to inter-annotator variability. Panel (c) shows that the relative F1 score depends sensitively on the number of annotators. Here, the number of annotators refers to the total number used in the analysis (6 annotators are used for the hold-out set while the number used in the ground truth set is varied). Panel (d) shows that the relative F1 score (where we used 6 annotators in the ground-truth set and 3 hold-out annotators) also gets skewed when annotators are allowed to voluntarily skip images (or equivalently not all annotators get images of equivalent difficulty). For panels (c) and (d) we have fixed the swap probability at 0.3.

#### 4.1.1    NAIVE EVALUATION

In this regime, we obtain each observed label by either selecting the true label (probability $1 - p$) or random sampling (probability $p$). As Fig. 1a illustrates, the degree to which model performance is underestimated increases monotonically and linearly with the label swap probability. The Naive Evaluation strategy is limited in its reliance on one label per sample. It has no way to evaluate the degree of label noise (a priori) nor a mechanism to compare model to annotator performance.

### 4.1.2 MAJORITY VOTE

In this regime, we assume we have 9 different annotators, each of whom selects the true label with probability $p$ or randomly samples a label with probability $1 - p$. The final observed label is obtained by taking the mode of the annotators' labels. Fig. 1b shows the effect of this aggregation on the observed accuracy. Majority Vote has an advantage over Naive Evaluation in that the narrowing gaps between models as the label noise is increased can, to some extent, be recovered by averaging multiple labels. Additionally, one can use the entropy of the label distribution to get a sense of inter-annotator variability. However, although Naive Evaluation and Majority Vote are sufficient for the task of ranking models (deciding which one is better), these approach do not indicate whether the model performance is better or worse than the average human performance.

### 4.1.3 RELATIVE F1

We additionally consider the method used in (Rajpurkar et al., 2018b; Hannun et al., 2019; Rajpurkar et al., 2018a). Here, a model and a set of held-out annotators are compared (using the F1 score) to a gold-standard ground truth set obtained using the majority-vote of multiple annotators. The model F1 score is then compared to the average F1 score of all the annotators in the hold-out set. As can be seen in Fig. 1c, the difference in the model and average annotator F1 scores, which we refer to as Relative F1, is dependent on the number of annotators used in the ground truth set (see Appendix C for intuition about this effect).

The Relative F1 score metric has another major shortcoming. As we discuss in Appendix C, in many applications annotators must be allowed to voluntarily skip samples when they feel that there is not enough information to provide an accurate label. Consequently, as shown in Fig. 1d a single annotator can bias the relative F1 score by simply annotating only the easier samples (on which there is high agreement among annotators) and skipping the hard ones (on which the agreement is low). Here we assigned to each image in the MNIST test set its own swap probability $p$, which ranges from lower values (i.e. images on which there is high agreement) to high values (i.e. images on which there is high disagreement). We find that as we increase the single annotator strictness (the fraction of the dataset they skip), the relative F1 score is correspondingly reduced. It should be noted that this same exact bias can arise when comparing the model and all annotators pairwise (as in the methods used in (Wei et al., 2019; Rajpurkar et al., 2017)) rather than to the majority vote.

### 4.1.4 DISCREPANCY RATIO

The discrepancy ratio introduces a number of improvements over the baselines. First, unlike Naive Evaluation and Majority Vote, the discrepancy ratio explicitly compares the model performance to the inter-annotator variability. As illustrated in Figs. 2a and 2b, when the swap probability is close to 0, the annotators are in almost perfect agreement and the discrepancy ratio is higher than 1 (for models other than model 1.0). When $p$ increases past a certain point, however, the annotators become more discrepant relative to each other than relative to the models and the discrepancy ratio falls below 1. **It is in this regime that the model performance can be said to be better than the average human performance.** Once $p$ approaches 1, the labels are completely random and the annotators are again as different from each other as they are from the model, yielding a discrepancy ratio of 1. Naturally, the discrepancy ratio for the ground truth curve (model 1.0) always lies below 1. We note that the large gap in performance between models in the low-noise regime represents an important and subtle feature of our approach: models that have similar performance as measured by the Naive Evaluation and Majority Vote methods can in fact be dramatically different when human performance is taken into account. As a specific example, consider models 1.0 and 0.9 at a label swap probability of 0.05. These models appear to only be different by $\sim 10\%$ when evaluated using accuracy metrics (Figs. 1a and 1b). However, as shown in Figs. 2a and 2b, model 0.9 is twice as far from annotators as they are from each other, while model 1.0 is twice as close, thus resulting in a discrepancy ratio performance gap of $\sim 400\%$.

Relative F1 *does* allow for evaluating the model by comparing it to annotator performance but the discrepancy ratio has several advantages. First, as illustrated in Fig. 2c, the discrepancy ratio is insensitive to the number of annotators making it a more robust metric to compare across different published studies. Second, the discrepancy ratio allows for a highly varying number of annotators per sample. This feature permits its use in scenarios where annotators have the ability to skip samples and makes it a more practically viable solution than Relative F1. The presence of strict annotators (who skip hard samples) has no effect on the discrepancy ratio (Fig. 2d).

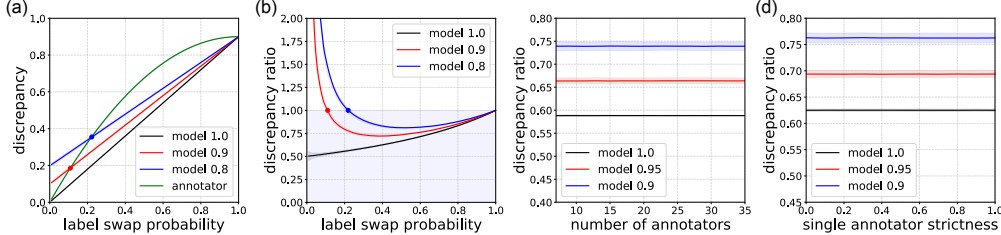

Figure 2: Comparison of the discrepancy ratio with baseline metrics on the MNIST test set. Curves show mean value and 2-standard deviation bounds obtained from multiple simulations of the experiment. Panel (a) shows the model discrepancy (for three models) and annotator discrepancy. The red and blue points occur at label swap probabilities of 0.1 * 10/9 and 0.2 * 10/9, respectively, which correspond precisely to when the accuracies of the simulated human annotators equal that of the models (0.9 and 0.8, respectively). The discrepancy ratios (Eq. 4) for the same three models are shown in (b). The blue and red points correspond to those shown in (a). The shaded region denotes the area where the model is superior to the average human performance. Panels (c) and (d) show that the discrepancy ratio is not sensitive to the number of annotators used, nor is subject to the strict annotator bias illustrated in Fig 1d. Here we have used the same parameters as in Figs 1c and 1d.

| Dataset | No. of Images | No. of Patients |
|---|---|---|
| Train | 60,411 | 1,915 |
| Validation | 5,996 | 200 |
| Test | 17,659 | 600 |

Table 2: Training and Evaluation Datasets for PLAX Measuraility Classification Task.

## 4.2 ECHOCARDIOGRAPHIC IMAGE CLASSIFICATION AND REGRESSION

An echocardiogram is a common medical test that uses high-frequency sound waves (ultrasound) to visualize the structure of the heart. In this work, we consider two important and ubiquitous tasks that are performed as part of virtually any echocardiogram (Lang et al., 2015).

**PLAX measurability classification**: The exam typically begins with acquiring a parasternal long axis (PLAX) view, which is used to get an overview of general cardiac function, as well as to make specific measurements (e.g. fractional shortening (Lang et al., 2015), presence of pericardial effusion, etc). Before any measurement can be performed, however, a clinician must first perform a binary classification task and determine whether the image in question is of sufficient quality to be used in the calculation (a similar binary classification screening step must be performed for most medical imaging tasks).

**LVEF regression**: The left ventricular ejection fraction (LVEF), which measures the fraction of blood the left ventricle expels with each heart cycle, is one of the key metrics used to quantify heart functionality. This quantity is often measured in the apical 4-chamber view of the heart by segmenting the left ventricular endocardial border at end-systole and end-diastole (see Appendix A).

Both tasks described above currently require the significant expertise of a trained sonographer. We have developed deep neural network models to automate these tasks. The datasets for these tasks are summarized in Tables 2 and 3. All data was collected using the Butterfly iQ ultrasound probe. Details of the data collection and annotations for both tasks are described in Appendix A.

| Dataset | No. of Images | No. of Patients |
|---|---|---|
| Train | 890,541 | 2,964 |
| Validation | 407,151 | 1,180 |
| Test | 149,992 | 322 |

Table 3: Training and Evaluation Datasets for LVEF Regression Task.

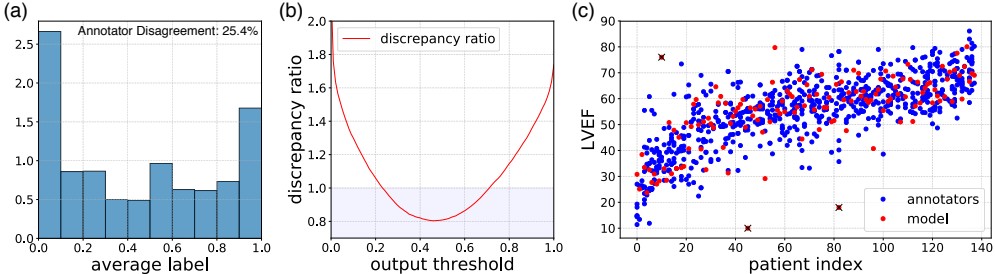

Figure 3: Panel (a) shows that human experts exhibit a high amount of disagreement on which images meet the threshold for measurability (i. e. with perfect agreement all labels would be 0 or 1). This plot displays the normalized histogram of the average labels on the PLAX measurability classification dataset. The discrepancy ratio on the test set as the output threshold of the network output is varied is shown in (b). Panel (c) shows the labels (blue) and model predictions (red) on the LVEF regression task test set, sorted by the mean LVEF label. Each patient was labeled by an average of 5 annotators. The black crosses signify the points on which the model predictions were artificially corrupted to create large errors.

### 4.2.1 EVALUATION USING DISCREPANCY RATIO

In order to explore the properties of the discrepancy ratio in a real-world setting, we first consider the PLAX measurability classification dataset. The neural network used for this task is described in Appendix D. When evaluating the performance of our network using Naive Evaluation (where we evaluate on a single annotator, chosen at random), we find that we achieve an accuracy of $77.8\%$, while the majority vote method gives an accuracy of $88.0\%$. These numbers are quite poor by the standards of most machine learning applications. Indeed, we might otherwise conclude that the model performs at a sub-standard level.

However, when we evaluate using the discrepancy ratio and consider the inter-annotator variability, a very different story emerges. As can be seen in the distribution of the average label ($y^i$ from Eq. 5) in Fig. 3a, there is significant disagreement ($\sim 25\%$ on average) among the annotators as to whether a given image meets the threshold of measurability. While this level of disagreement may seem high, it is typical of that found in the medical applications literature (Wei et al., 2019; Hannun et al., 2019).

Fig. 3b shows the discrepancy ratio as a function of the threshold of the network output. Contrary to the impression suggested by using the conventional metrics, we find that for a wide range of thresholds the model dramatically exceeds the average human performance (as defined by $\Delta = 1$). Table 4 shows the discrepancy ratio, with $95\%$ confidence intervals, for the model as compared to those achieved by the individual annotators on the test set. Here we find that the discrepancy ratio has another nice property: it can be used to evaluate the performance of individual annotators. We find that there is a significant spread in the annotator performance and that our model outperforms even the best annotator in the dataset.

### 4.2.2 DEPENDENCE ON $\delta$

As described in Section 2, the discrepancy ratio can be thought of as a "meta-metric" that evaluates the model performance for any desired measure of agreement $\delta$. Here we use the LVEF regression dataset to explore this dependence on $\delta$. The neural network we used is described in Appendix D. We consider three choices of $\delta$: the absolute difference: $\delta = |y - y'|$, squared difference: $\delta = (y - y')^2$ and the hinge loss: $\delta = \max(0, \lambda - |y - y'|)$, where a penalty is incurred only past a minimum deviation threshold [1]. As with the PLAX measurability classification task, we find that our model exceeds the performance of even the best annotator in our dataset on all three measures of agreement, obtaining discrepancy ratios of 0.89, 0.78 and 0.67 respectively. In order to demonstrate the dependence on $\delta$, we then artificially corrupt the model so that it occasionally makes errors that are far outside the range of those made by humans (Fig. 3c). How does the model fare with respect to human performance? It depends on the type of errors one cares most about. When we set $\delta$ to be the absolute difference, the discrepancy ratio is 0.95 and one may conclude that the model is performing well. If the squared difference is used for $\delta$ instead, the outliers receive higher cost and the discrepancy ratio becomes

---

[1]We used $\lambda = 15$ as a clinically relevant measure of a significant Ejection Fraction error.

|  | Δ (95% CI) |
|---|---|
| A1 | 1.160 (1.117 to 1.213) |
| A2 | 0.904 (0.875 to 0.932) |
| A3 | 0.907 (0.858 to 0.964) |
| A4 | 0.922 (0.896 to 0.946) |
| A5 | 1.105 (1.052 to 1.161) |
| A6 | 0.959 (0.914 to 1.010) |
| A7 | 0.952 (0.923 to 0.984) |
| A8 | 1.131 (1.065 to 1.215) |
| A9 | 0.932 (0.897 to 0.967) |
| A10 | 1.030 (0.983 to 1.073) |
| A11 | 1.009 (0.973 to 1.049) |
| A12 | 1.069 (1.029 to 1.112) |
| Model | **0.805 (0.786 to 0.820)** |

Table 4: The discrepancy ratio $\Delta$ for the model and annotators (who have labeled at least 100 patients) on the PLAX measurability test set. Here each annotator was individually taken out of the dataset and treated as if they were the model for the purpose of calculating their discrepancy ratio. The confidence intervals were obtained using the bootstrap method where the sampling was performed by patient. We used the validation set to choose the threshold corresponding to the lowest discrepancy ratio (as in Fig. 3b).

1.02. When a hinge-loss is used for $\delta$, where deviations below $\lambda$ are considered acceptable, we observe that the performance is worse still (1.08).

## 5 LIMITATIONS

While our approach provides for a simple and general way of quantifying model performance in the presence of label noise, it has several limitations. First, although the flexibility in choosing $\delta$ allows us to examine various aspects of the model performance (illustrated in the example above), our metric needs to be applied with caution since it is only as good as the underlying measure of agreement. Second, in situations when annotators are allowed to skip samples they feel unsure about, samples that get fewer labels may tend to be more ambiguous. Since the discrepancy ratio is undefined for samples with fewer than two labels, this can cause the discrepancy ratio to overweight easier examples. Consequently, one should ensure that all samples in the domain of interest are labeled by at least two annotators. Finally, the discrepancy ratio can yield misleading results when the labels are provided by poor annotators, though this is also true of any metric (e.g. the relative F1) that attempts to normalize model performance to account for annotator disagreement. To mitigate this, the annotators should be properly vetted as skilled for the task (see e.g. Hovy & Lavid (2010) or our vetting procedure in Appendix A) and allowed to skip when they are unsure of the label. In some situations annotators may adhere to several discrete "schools of thought" and hence their annotations may be more similar within each group than between groups. In these situations it may make sense to evaluate the discrepancy ratio independently within each group.

## 6 CONCLUSION

We present a unified and principled approach to evaluating machine learning models when humans disagree on the truth. Our metric, the discrepancy ratio, is highly intuitive and can be adapted to virtually any task. We have shown that the discrepancy ratio has numerous desirable properties including insensitivity to annotator count and insensitivity to annotator/image overlap. At the same time, we have evaluated the shortcomings of alternative implementations and show that our method has numerous advantages that make it a convenient standard by which models can be evaluated in the presence of ambiguous ground truth labels. Finally, to demonstrate our method we have trained deep neural networks on two real-world tasks from medical imaging and have used our metric to show that we achieve performance better than even the best annotators in our datasets.

ACKNOWLEDGMENTS

The authors would like to thank Gal Koplewitz for his input and several helpful discussions.

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

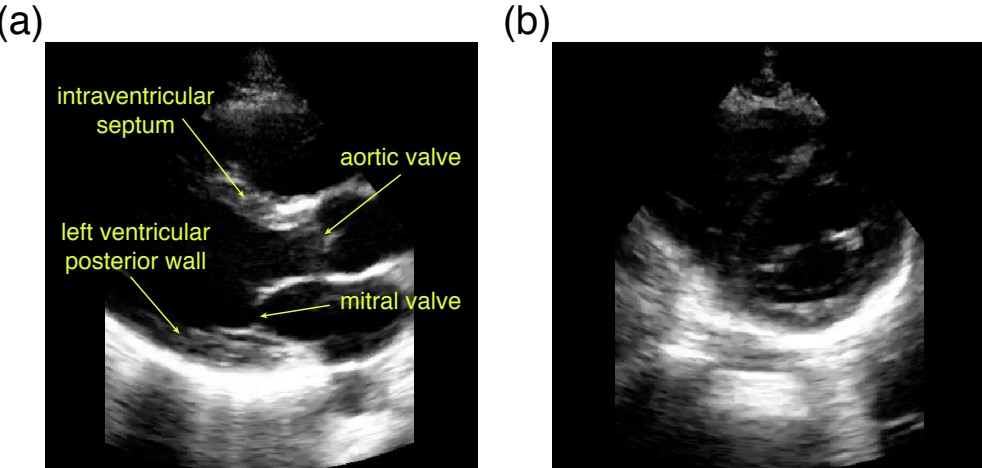

Figure S1: Examples of images deemed measurable (a) and not measurable (b) by majority vote of annotators. Several important features characteristic of a measurable image are noted in (a). These same features are obscured or entirely absent in (b).

## A  DATA COLLECTION, DATASETS AND ANNOTATIONS

In this section we describe the data collection, dataset splits and annotation procedure for the tasks presented in Section 4.2 of the main text.

### A.1  DATA COLLECTION

For model development and evaluation, images were collected in several hospitals in the United States on patients presenting for a standard echocardiogram exam. All data was collected using the Butterfly iQ probe. For the PLAX measurability classification dataset we have additionally included images taken on healthy adults outside the hospital setting (approximately 10% of the dataset).

### A.2  PLAX MEASURABILITY CLASSIFICATION TASK

In order to collect labels, echocardiogram images were presented to 14 professional, US-licensed sonographers. The images in the test set received an average of 9 labels per image. The sonographers were given the following task:

---

Mark as **MEASURABLE** the frames that satisfy ALL of the following criteria:

- The frame shows a parasternal long axis (PLAX) view of diagnostic quality.

- The frame can be used for a measurement of the left ventricular internal diameter (LVID), as shown in the diagram. Note that the LVID can be measurable irrespective of where the heart is in its cycle.

Mark all other frames as **NOT MEASURABLE**.

---

The above prompt refers to a diagram showing a canonical image of a PLAX view. Examples of images deemed measurable and not measurable by the majority of annotators are shown in Fig. S1.

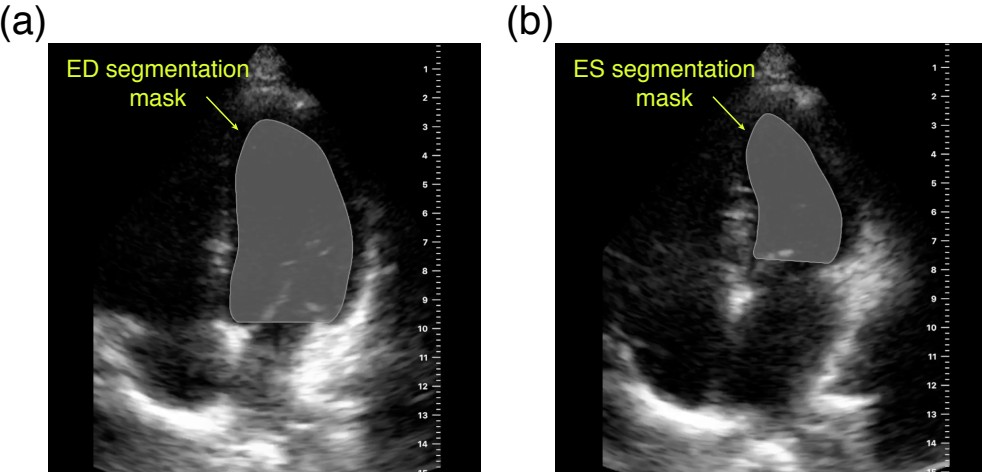

Figure S2: An example LVEF calculation in the apical 4-chamber view. Panel (a) shows the end-diastole frame with the overlaid segmentation mask while panel (b) shows the corresponding end-systole frame within the same heart cycle. The calculated LVEF from these segmentation masks is 66%

## A.3   LVEF REGRESSION TASK

Here we collected three separate types of labels:

1. **Segmentation Masks**: annotators were asked to segment the endocardial border of the left ventricle.

2. **Measurability**: annotators were asked to classify frames into measurable/non-measurable classes using a procedure virtually identical to that used in the PLAX measurability classification task above.

3. **LVEF labels**: annotators were asked to provide LVEF labels by segmenting the endocardial border at the end-systole and end-diastole frames.

Since the model predicts segmentations and measurability scores only as intermediate states (see Appendix D for more details), they are not relevant for the evaluation framework presented in this work and we do not consider these predictions further. Instead we focus on the LVEF regression predictions. For this task, short echocardiogram video clips containing several heart cycles were presented to 19 professional, US-licensed sonographers. The videos in the test set received an average of 5 labels per video. The sonographers were given the following task:

> Apical Endocardial contours should be labeled according to ASE guidelines:
>
> - Identify a heart cycle within the cine.
>
> - Navigate to the ED frame and trace the endocardial border of the left ventricle.
>
> - Navigate to the ES frame and trace the endocardial border of the left ventricle.
>
> - Make sure the ED frame is to the left of the ES frame but part of the same heart cycle.
>
> - Observe the calculated EF just above the cine. If this EF score looks correct, hit Submit. Otherwise, adjust the ED and ES frames accordingly.

The above prompt refers to "ASE guidelines", which are published consensus recommendations and can be found at Lang et al. (2015). An example of a LVEF prediction is shown in Fig S2, where the model segments the end-systole and end-diastole frames and calculates the LVEF via the Modified Simpson's Method (Lang et al., 2015).

### A.4 ANNOTATOR INCLUSION CRITERIA

We relied on two senior cardiac sonographers (each having more than 15 years of experience) to source annotators. For both tasks, each annotator was interviewed and given an onboarding assignment. Annotators that were deemed by our sonographers to comply with ASE standards were then cleared to label images. During the first several weeks of label collection, our sonographers conducted several detailed reviews of each annotator's images. Examples in which our sonographers disagreed with the assessment of the annotator were discussed with the annotator in live sessions to ensure that the disagreements resulted in genuine differences of medical opinion, rather than annotation errors. In addition, the disagreement metrics for each annotator (as calculated Section 4.2.1) were tracked over time and were used to schedule additional review sessions with the annotators, as necessary.

## B MPD AND ANNOTATOR DISCREPANCY USING SQUARED-ERROR AGREEMENT FUNCTION

As a preliminary result that is used in Section 3.3, we show that given any two sets of labels $\mathcal{Y} = \{y_1, y_2, ..., y_n\}$ and $\mathcal{Y}' = \{y_1', y_2', ..., y_{n'}'\}$ the MPD is given by the expression

$$\psi(\mathcal{Y}, \mathcal{Y}') = (y - y')^2 + \sigma_y^2 + \sigma_{y'}^2,$$

where $y = \frac{1}{n} \sum_{k=1}^{n} y_k$ and $\sigma_y^2 = \left(\frac{1}{n} \sum_{k=1}^{n} y_k^2\right) - y^2$ are the mean and variance of $y_k$, respectively.

*Proof.*

$$\psi(\mathcal{Y}, \mathcal{Y}') = \sum_{k=1}^{n} \sum_{k'=1}^{n'} \frac{(y_k - y_{k'}')^2}{nn'}$$

$$= \frac{1}{nn'} \left[ n' \sum_{k=1}^{n} y_k^2 + n \sum_{k'=1}^{n'} y_{k'}^2 - 2 \sum_{k=1}^{n} y_k \sum_{k'=1}^{n'} y_{k'} \right]$$

$$= \frac{1}{nn'} \left[ n'n(\sigma_y^2 + y^2) + nn'(\sigma_{y'}^2 + y'^2) - 2nn'yy' \right]$$

$$= (y - y')^2 + \sigma_y^2 + \sigma_{y'}^2.$$

$\square$

As a special case, when the labels can only take on the values 0 and 1, the variance is fully determined by the mean:

$$\sigma_y^2 = y - y^2,$$

and the MPD simplifies to

$$\psi(\mathcal{Y}, \mathcal{Y}') = y + y' - 2yy'.$$

Using this result and the definition in Eq. 2, the annotator discrepancy for a single sample is then given by:

$$\alpha^i = \sum_{\substack{j,j'=1 \\ j \neq j'}}^{A^i} \frac{y_j^i + y_{j'}^i - 2y_j^i y_{j'}^i}{A^i(A^i - 1)}$$

$$= \frac{1}{A^i(A^i - 1)} \left[ 2(A^i - 1) \sum_{j=1}^{A^i} y_j^i - 2 \sum_{\substack{j,j'=1 \\ j \neq j'}}^{A^i} y_j^i y_{j'}^i \right]$$

$$= 2y^i(1 - y^i) + \frac{2\sigma_{y^i}^2}{A^i - 1}.$$

where $\sigma_{y^i}^2$ is the variance of the labels $y^i$. Here we have used the fact that

$$\sum_{\substack{j,j'=1 \\ j \neq j'}}^{A^i} y_j^i y_{j'}^i = (A^i y^i)^2 - \sum_{j=1}^{A^i} (y_j^i)^2.$$

For the special case where each annotator provided no more than a single label per image, the above relationship between the mean and variance can be used to further simplify the annotator discrepancy to

$$\alpha^i = \frac{2A^i}{A^i - 1} \sigma_{y^i}^2.$$

## C  ALTERNATIVE FORMULATIONS OF THE DISCREPANCY RATIO

In constructing the discrepancy ratio metric, we made several seemingly arbitrary design choices. In this section we discuss the reasons for these decisions and why the alternative formulations are problematic.

### C.1  THE ORDER OF THE AVERAGES OVER ANNOTATIONS AND SAMPLES

Consider an alternative definition for the model discrepancy, where we swap the order of the averages over samples and annotations:

$$\mu' = \sum_{j=1}^{A} \sum_{\substack{i=1 \\ \mathcal{Y}_j^i \neq \varnothing}}^{N} \frac{\psi(m^i, \mathcal{Y}_j^i)}{NA}.$$

Here $A$ is the total number of annotators in the dataset and the sum over samples has to go over only those $i$ for which the $j$th annotator has labeled it at least once (i.e. $\mathcal{Y}_j^i \neq \varnothing$). Conceptually, this expression calculates a discrepancy score for each model/annotator pair, and then averages them together to arrive at an aggregated measure of agreement.

An analogous alternative definition can be constructed for the annotator discrepancy:

$$\alpha' = \sum_{\substack{j,j'=1 \\ j \neq j'}}^{A} \sum_{\substack{i=1 \\ \mathcal{Y}_j^i, \mathcal{Y}_{j'}^i \neq \varnothing}}^{N} \frac{\psi(\mathcal{Y}_j^i, \mathcal{Y}_{j'}^i)}{NA(A-1)}.$$

As in the case of the model discrepancy, this expression computes a discrepancy for each annotator pair and then averages all the pairs to yield an aggregate measure of annotator agreement. The methods presented in Wei et al. (2019); Rajpurkar et al. (2017) use essentially these definitions, though they do not compare them using a ratio.

When all annotators label all samples, these definitions are perfectly valid, and are in fact equivalent to $\mu$ and $\alpha$. However, in most real-world scenarios, data come from disparate sources (e.g. different hospitals with different annotators) and it is not practical to ensure that the exact same set of annotators labeled all the samples. Worse still, in many domains it is necessary to allow human annotators to voluntarily skip samples when they feel there is not enough information to provide an accurate label. In such a scenario, formulating the model and annotator discrepancies using the alternative definitions above allows an annotator to improve their own discrepancy score by simply being strict and only annotating "easy" samples. This will generally result in a bias being introduced into the discrepancy ratio. To illustrate this point, consider the following toy example:

**Example C.1** A dataset contains one "easy" image (on which the MPD between any two annotators is $\delta_e$) and one "hard" image (on which the MPD between any two annotators is $\delta_h$). Similarly, a machine learning model gets an MPD of $\delta_e'$ on the easy image and $\delta_h'$ on the hard image. Three annotators (Alice, Bob and Charlie) are asked to label the data. Alice and Bob label both images, but Charlie wants to improve his discrepancy and only labels the easy image. Using the original definitions of $\alpha$ and $\mu$, the discrepancy ratio is

$$\Delta = \frac{\mu}{\alpha} = \frac{(\delta_e' + \delta_h')/2}{(\delta_e + \delta_h)/2} = \frac{\delta_e' + \delta_h'}{\delta_e + \delta_h}.$$

Note that the value of $\Delta$ would have been exactly the same without Charlie. On the other hand, using the alternative definition the discrepancy ratio becomes biased and is given by

$$\Delta' = \frac{\mu'}{\alpha'} = \frac{(2(\delta_e' + \delta_h')/2 + \delta_e')/3}{((\delta_e + \delta_h)/2 + 2\delta_e)/3} = \frac{2\delta_e' + \delta_h'}{(5\delta_e + \delta_h)/2}.$$

In the special case where the annotators and model are in perfect agreement on the easy image ($\delta_e = \delta_e' = 0$), Charlie was able to double the discrepancy ratio by simply skipping the difficult image:

$$\Delta' = \frac{2\delta_h'}{\delta_h} = 2\Delta.$$

It should be noted that even if we weigh the average of annotators by the inverse of the number of annotations they provided, the bias does *not* go away, whereas the discrepancy ratio, as defined in the main text, is completely insensitive to this effect. The effect illustrated in the example above is not limited to the presence of an adversarial annotator, but can creep into the data whenever there exist any systematic differences in how the data got partitioned among annotators, which in practice may be challenging or even impossible to control.

## C.2 Computing Pairwise Deviations vs. Comparisons to the Mean

In the definition of the annotator discrepancy, we compute the MPD over all pairs of annotators. A seemingly valid alternative would be to instead compare each annotator to the mean of all the other annotations. In this alternative definition, the annotator discrepancy is given by the expression

$$\alpha'' = \sum_{i=1}^{N} \sum_{j=1}^{A^i} \frac{\psi(\mathcal{Y}_j^i, \omega_j^i)}{N A^i},$$

where

$$\omega_j^i = \sum_{\substack{j'=1 \\ j' \neq j}}^{A^i} \frac{y_{j'}^i}{A^i - 1}.$$

Here, $\omega_j^i$ is a scalar that represents the average of all annotations for the $i$th sample, leaving out the $j$th annotator. We can analogously construct an alternative definition for the model discrepancy:

$$\mu'' = \sum_{i=1}^{N} \sum_{j=1}^{A^i} \frac{\psi(m^i, \omega_j^i)}{N A^i}.$$

Note that we cannot simply compare the model predictions to the average of *all* the annotations since then the number of annotators that go into the average for $\mu''$ would be one higher than that used in $\alpha''$, which would artificially lower the discrepancy ratio. The methods presented in Hannun et al. (2019); Rajpurkar et al. (2018a;b) are similar to this averaging approach.

While $\mu''$ and $\alpha''$ may seem like valid formulations for the model and annotator discrepancies, they lead to a major problem: *The discrepancy ratio becomes dependent on how many annotators are used in the study.*

To see why this is, notice that $\omega_j^i$ is an average over $A^i$ annotators and will generally scale as $1/\sqrt{A^i}$. This implies that both $\mu''$ and $\alpha''$ will get smaller as the number of annotators is increased. They will generally scale at different rates, however, depending on the distribution of the model predictions and annotator labels, and therefore will make the ratio dependent on the number of annotators (for a specific example, see Section 4 of the main text).

## D  Algorithm Development and Training

The deep learning algorithm used for the PLAX measurability classification task is a 10-layer convolutional neural network, where each 2-layer block subsamples the inputs by a factor of 2. After the final layer we apply a global average pooling, followed by a sigmoid function that is thresholded at a particular level (see Section 4). We trained the network *de novo* using the Adam optimizer with a learning rate of $0.00005$ and a mini batch size of 32.

The algorithm used for estimating the LVEF employs a U-Net style network tasked with segmenting the left ventricular endocardial border. The network consists of a 6-layer encoder followed by 6-layer decoder where downsampling and upsampling by a factor of 2 occur at each layer, respectively. We trained the network *de novo* using the Adam optimizer with a learning rate of $0.0001$ and mini batch size of 64. The network contains an additional output head that predicts the measurability of the input image. The per-frame segmentations and measurability values are then used to find the end-systole and end-diastole frames, which are in turn used to calculate the LVEF via the Modified Simpson's Method (Lang et al., 2015). The per-frame measurability scores are aggregated to produce a quality score for each LVEF. The model only outputs LVEF predictions for which the predicted quality is

above a predefined threshold.

## E    EXTENSIONS OF THE DISCREPANCY RATIO

When there exists an imbalance of labels in the dataset, the discrepancy ratio metric can be stratified to give a rebalanced measure of agreement.

For example, in a classification task where the class labels are imbalanced, we can define the mean-per-class discrepancy ratio by separately computing $\Delta$ for each class and averaging the results together (in this case, the stratification into different classes can be determined by majority vote).

