# OpenReview forum: "Discrepancy Ratio: Evaluating Model Performance When Even Experts Disagree on the Truth"
_ICLR.cc/2020/Conference — Accept (Poster)_

### Official Review · AnonReviewer3 · 2019-10-21
**Official Blind Review #3**

**Rating:** 8

**Review:**

Summary: This paper proposes a novel metric "discrepancy ratio" for evaluating the performance of a model where the ground truth for each data point comes from many expert-yet-imperfect annotators. The authors suggested that this problem has remained largely unexplored. The proposed metric is intuitive and is easy-to-use. The authors suggested that this metric can be used for many applications. For example, to evaluate if a model is better than the annotators on average, to evaluate the annotator, to compare between models.

========================================================
Comments on clarity:

The writing is good overall and I am able to understand the contribution and the key idea of this paper.

Questions and comments on clarity:

1. While MPD is defined with an argument (Y, Y'), the definitions annotator discrepancy, model discrepancy and discrepancy ratio ignore the inputs of the function. It might be better to keep the arguments of the function for clarity. When people want to adopt the discrepancy ratio in their work and there are many ratios to discuss, then the proposed definition can be used conveniently.

2. In Eq. (5) \sigma seems to be undefined. I believe it is a variance but it is kinder to readers to explicitly state it (I found the authors explicitly stated it in Appendix B).

3. In Eq. (8), is it intentional to have $i$ exists over m and y as written in the paper?

4. Does the discussion of the simplicity of the discrepancy ratio hold for other \delta, i.e., not the squared loss?

5. In Section 4.1 (experiment with MNIST), did the authors binarize the data or simply discuss the multi-class problem? Because this follows section 3.3 which dedicates the whole page to discuss binary classification. It may be a bit sudden to move to multi-class classification and an additional sentence to clarify this might help.

6. I could not understand what is "single annotator strictness in Figures 1d and 2d. It is not explained if I did not miss it.

7. I think we should not resize the figure/table caption. The text size of captions are much smaller than the main text.

8. Is PLAX binary classification? (I understood as yes from Figure (3a) but I think it's also nice to clarify it to avoid confusion (also for MNIST)).

9. What is the y-axis for figure 3a?

10. I think reporting the performance of the mean-per-class discrepancy ratio without defining it in the main text (it is defined in a natural language (English) in Appendix E) makes the main body of the paper not self-contained. I strongly suggest to either (1) remove mean-per-class discrepancy ratio entirely from the main body or (2) define it mathematically in the main body. Also, I don't see reporting mean-per-class discrepancy ratio adds values to a paper for current experiments because the reported mean-per-class ratio is very similar to that of discrepancy ratio (Figures 3b (almost overlap) and Table 2 (same trend and the values are very close to the discrepancy ratio for all cases)).

11. Can we use F1 to evaluate whether a model is better than the average human performance?

Minor points on clarity, which may just be a personal preference of a reviewer:

1. The authors may consider using \begin{definition} when defining three key values: the mean pairwise deviation (MPD), annotator discrepancy, and model discrepancy. So the readers can easily look up when getting lost in definitions. And it may help to make a definition more precise (e.g., given label Y, data points, MPD is defined as follows).

2. Eqs (1), (2), (3) may be easier to understand if we put a denominator out of the summation as much as we can. For example, for Eq. 2, N can be outside of the two sums, A^i(A^i-1) can be between the two sums.

========================================================
Significance:
The problem that this paper addresses is highly important. I like the idea and I am convinced that discrepancy ratio is potentially useful for the community. And it also opens many possibilities to analyze the discrepancy ratio and to find potential applications for it.

========================================================
Other comments:

On the limitation:
I am satisfied that the authors clearly discuss the limitations of the proposed method and I agree with the discussion.

On the experiments:
1. Why we use model 1.0, 0.95, 0.9 for Figures 1c, 1d, 2c, 2d but 1.0, 0.9, 0.8 for Figures 1a, 1b, 2a, 2b? Perhaps full results with many models should also be reported for completeness.

2. Isn't it more natural to show a graph that has x-axis as label swap probability in the same figure for comparison? For example, Figure 1 only report performance with respect to each measure (Naive Evaluation, Majority Vote, Relative F1, Discrepancy ratio) with respect to label swap probability. We can fix the number of annotators to 9 for relative F1 and discrepancy ratio, to validate whether it also becomes increasingly difficult to differentiate model improvements or not for them. Then after suggesting that Naive evaluation is not good and other metrics look fine. Majority vote cannot evaluate that the model is better than human average performance, which we want to know. Then Figure 2 highlights the contribution of the discrepancy ratio and shows that it outperforms relative F1 in some perspectives.

3. For the MNIST benchmark dataset, it is possible to show the performance on the test ground truth too. And I think it is interesting to compare different models with the same label noise rate and confirm that discrepancy ratio is successful to evaluate the best model with respect to the ground truth accuracy. It would be very impressive to see if the best model with respect to the discrepancy ratio can successfully be the best also on the ground truth test set.

On space usage:
I feel there are many important things to be included in the main body but it is instead in the appendix. On the other hand, I think there are several parts that can be shortened in the main body. And it seems that the binary classification example (3.3) also consumed a lot of space. The derivation may be omitted and make it a Theorem and then let's suggest the readers to read the appendix for the proof if they are interested. Then we can have more space to add important components in the main body of the paper.

========================================================
Decision:
Although there are issues on the clarity of the paper, I believe these issues are not too difficult to fix in the final version. The idea is interesting and potentially impactful. For these reasons, I vote a weak accept for this paper.


Update: I would like to thank the authors for the clarification. I have read the rebuttal.
I read through the updated version of the paper. I believe that although there is still room for improvement, the contribution of this paper is sufficient as an important step towards solving this difficult yet highly relevant problem. This paper also stimulates several future directions. The authors clearly stated the limitations of the paper, which is highly useful for researchers to improve this work or find a better solution.  The clarity of the paper is also improved. For these reasons, I increased my score from weak accept to accept.

**Experience Assessment:**

I do not know much about this area.

**Review Assessment: Checking Correctness Of Derivations And Theory:**

I assessed the sensibility of the derivations and theory.

**Review Assessment: Checking Correctness Of Experiments:**

I carefully checked the experiments.

**Review Assessment: Thoroughness In Paper Reading:**

I read the paper thoroughly.

---

> ### Author Response · Authors · 2019-11-11
> **Response to Official Blind Review #3 (1/2)**
>
> We thank the reviewer for their positive review of our work. We specifically appreciate their highlighting the fact that “The problem that this paper addresses is highly important”, that our proposed solution is “intuitive and is easy-to-use” and that the reviewer is “convinced that discrepancy ratio is potentially useful for the community.” We have addressed in detail all comments and suggestions made by the referee and feel that this has substantially strengthened our paper. We hope that the resulting paper (uploaded) now merits clear acceptance. We strongly feel that our work is of critical importance in allowing the community to evaluate models in real-world settings where we have no access to unambiguous ground truth labels. To the best of our knowledge, our work represents the only approach in the literature that provides a robust, task-independent, interpretable metric to answer the crucial question of how a machine learning model performs at a task relative to humans.
>
> “1. While MPD is defined with an argument (Y, Y'), the definitions annotator discrepancy, model discrepancy and discrepancy ratio ignore the inputs of the function. It might be better to keep the arguments of the function for clarity. When people want to adopt the discrepancy ratio in their work and there are many ratios to discuss, then the proposed definition can be used conveniently.“
>
> This has been added to the revised paper.
>
> “2. In Eq. (5) \sigma seems to be undefined. I believe it is a variance but it is kinder to readers to explicitly state it (I found the authors explicitly stated it in Appendix B).”
>
> Good catch. We explicitly note that \sigma^2 is variance in the revised version of the paper.
>
> “3. In Eq. (8), is it intentional to have exists over m and y as written in the paper?”
>
> Yes, the subscript i refers to the sample index and the covariance is taken over this index. In order to avoid confusion this has been clarified in the revised version.
>
> “4. Does the discussion of the simplicity of the discrepancy ratio hold for other \delta, i.e., not the squared loss?”
>
> Yes, the interpretation of the discrepancy ratio is exactly the same for any measure of agreement (delta), as described in the last paragraph of Section 3.1. To summarize, the discrepancy ratio can be thought of as a “meta-metric” that measures how the model performs relative to humans, as compared to how they perform relative to each other for any measure of agreement. We elaborate further on this point in Section. 4.2.2.
>
> “5. In Section 4.1 (experiment with MNIST), did the authors binarize the data or simply discuss the multi-class problem? Because this follows section 3.3 which dedicates the whole page to discuss binary classification. It may be a bit sudden to move to multi-class classification and an additional sentence to clarify this might help.”
>
> We completely agree with the reviewer that this point was not clear in our initial submission. To clarify, in Section 4.1 we simply calculate the accuracy using the MNIST labels without binarizing. This has been clarified in the revised version.
>
> “6. I could not understand what is "single annotator strictness in Figures 1d and 2d. It is not explained if I did not miss it.”
>
> In Section 4.1.3, we state that “We find that as we increase the strictness of a single annotator (the fraction of the dataset they skip), the relative F1 score is correspondingly reduced.” The reviewer is absolutely correct that the exact phrase “single annotator strictness”, which refers to the statement above was never defined. This has been fixed in the revised paper.
>
> “7. I think we should not resize the figure/table caption. The text size of captions are much smaller than the main text.”
>
> This has been fixed.
>
> “8. Is PLAX binary classification? (I understood as yes from Figure (3a) but I think it's also nice to clarify it to avoid confusion (also for MNIST)).”
>
> Yes, the PLAX task is binary classification. This has been clarified in the revised paper.
>
> “9. What is the y-axis for figure 3a?”
>
> Figure 3a is a normalized histogram of the average labels on the PLAX classification dataset. This has been clarified in the caption.

---

> > ### Author Response · Authors · 2019-11-11
> > **Response to Official Blind Review #3 (2/2)**
> >
> > “10. I think reporting the performance of the mean-per-class discrepancy ratio without defining it in the main text (it is defined in a natural language (English) in Appendix E) makes the main body of the paper not self-contained.”
> >
> > We have removed the reporting of the mean-per-class discrepancy ratio from the main text of the paper. We have retained it in Appendix E since it may still be a useful extension of the discrepancy ratio in cases where the dataset is highly imbalanced.
> >
> > “11. Can we use F1 to evaluate whether a model is better than the average human performance?”
> >
> > Yes, as we state in the second paragraph of Section 4.1.4 that “Relative F1 does allow for evaluating the model by comparing it to annotator performance but the discrepancy ratio has several advantages”. The advantages are subsequently summarized.
> >
> > “1. The authors may consider using \begin{definition} when defining three key values: the mean pairwise deviation (MPD), annotator discrepancy, and model discrepancy. So the readers can easily look up when getting lost in definitions. And it may help to make a definition more precise (e.g., given label Y, data points, MPD is defined as follows).”
> >
> > We thank the reviewer for this suggestion. This will be changed in the final version of the paper.
> >
> > “2. Eqs (1), (2), (3) may be easier to understand if we put a denominator out of the summation as much as we can. For example, for Eq. 2, N can be outside of the two sums, A^i(A^i-1) can be between the two sums.“
> >
> > This will be changed in the final version of the paper.
> >
> > “On the experiments:
> > 1. Why we use model 1.0, 0.95, 0.9 for Figures 1c, 1d, 2c, 2d but 1.0, 0.9, 0.8 for Figures 1a, 1b, 2a, 2b? Perhaps full results with many models should also be reported for completeness.”
> >
> > The reason for this was purely aesthetic: in Figures 1c and 1d the curves for models 1.0, 0.95 and 0.9 are closer together, thus making it easier to see the dependence on the number of annotators and the single annotator strictness.
> >
> > “2. Isn't it more natural to show a graph that has x-axis as label swap probability in the same figure for comparison? For example? For example, Figure 1 only report performance with respect to each measure (Naive Evaluation, Majority Vote, Relative F1, Discrepancy ratio) with respect to label swap probability. We can fix the number of annotators to 9 for relative F1 and discrepancy ratio, to validate whether it also becomes increasingly difficult to differentiate model improvements or not for them. Then after suggesting that Naive evaluation is not good and other metrics look fine. Majority vote cannot evaluate that the model is better than human average performance, which we want to know.”
> >
> > We would like to ask for clarification from the reviewer on this comment. Is the reviewer asking for an additional panel to demonstrate how the Relative F1 score scales with the label swap probability?
> >
> > “3. For the MNIST benchmark dataset, it is possible to show the performance on the test ground truth too. And I think it is interesting to compare different models with the same label noise rate and confirm that discrepancy ratio is successful to evaluate the best model with respect to the ground truth accuracy. It would be very impressive to see if the best model with respect to the discrepancy ratio can successfully be the best also on the ground truth test set.”
> >
> > For the MNIST dataset, the performance on the test ground truth data corresponds to a label swap probability of 0. Thus this information is already included in Figures 1 and 2. The three models presented in each panel on these plots are also all evaluated with respect to the same label noise. Specifically, each point on the x-axis corresponds to a specific level of label noise. The discrepancy ratio is indeed successful at finding the best model - notice that in Figure 2b, the best model (Model 1.0) has the lower discrepancy ratio for all levels of label noise, indicating that it performs best as compared to annotators.
> >
> > “On space usage:
> > I feel there are many important things to be included in the main body but it is instead in the appendix. On the other hand, I think there are several parts that can be shortened in the main body. And it seems that the binary classification example (3.3) also consumed a lot of space. The derivation may be omitted and make it a Theorem and then let's suggest the readers to read the appendix for the proof if they are interested. Then we can have more space to add important components in the main body of the paper.”
> >
> > As requested by the reviewer, some of the details of the derivation of the example in Section 3.3 has been moved to Appendix B.
> >
> > In the final version of the paper we will move the following information from the appendices to the main text:
> > 1). Some details on the model training and architecture presented in Appendix D.
> > 2). Tables S1 and S2 detailing the dataset sizes for the PLAX and LVEF tasks.

---

> > > ### Comment · AnonReviewer3 · 2019-11-21
> > > **Clarification of my comment**
> > >
> > > The authors asked me that:
> > > "We would like to ask for clarification from the reviewer on this comment. Is the reviewer asking for an additional panel to demonstrate how the Relative F1 score scales with the label swap probability?"
> > >
> > > The short answer is yes. If I understand correctly, there is no F1 score scales with the label swap probability. So it may be nice to add it for completeness if possible. Nevertheless, if the authors believe that this is not important because we do not use this information to interpret the result or highlighting the importance of the relative F1 or the discrepancy ratio. Then I believe it is acceptable to not include it.
> > >
> > > I read through the updated paper and found that the clarity is improved. I increased my score to accept.

---

### Official Review · AnonReviewer1 · 2019-10-21
**Official Blind Review #1**

**Rating:** 6

**Review:**

This paper proposed to evaluate model performance when the ground truth labels were not available and noisy labels provided by multiple uncertain experts were provided instead. The proposed evaluation metric, called discrepancy ratio, is defined as the ratio between the average model-annotator discrepancy and the average annotator-annotator discrepancy. It can be applied to compare 1) the relative performance of different models; and 2) the relative performance of average annotators and the model. Experimental results on the MNIST classification task showed the proposed metric can compare model performance under different noise levels and is robust w.r.t. the number of annotators and a single annotator's strictness. Experimental results on a real-world medical image classification was also presented.

This work is well motivated. In real-world applications, such as medical image classifications, obtaining ground-truth labels is difficult or even infeasible, only multiple noisy labels are available to evaluate the model performance. Therefore, the setup of this problem seems novel (besides Rajpurkar et al. 2018).

In the experimental results on MNIST classification, for Naive Evaluation and Majority Vote, as the author pointed out, the gap between three models (model 1.0, model 0.9, model 0.8) becomes narrower when the label swap probability increases, indicating it's becoming more difficult to differentiate model improvements. However, this is also true for the proposed discrepancy ratio, as is shown in Figure 2(b). Therefore, the discrepancy ratio didn't overcome those "drawbacks" associated with these two baselines, correct? In other words, in the high-noise regime, it would be natural to observe narrower gaps between those models.

In MNIST classification, the performances gap of those three models evaluated by Naive Evaluation and Majority Vote didn't decrease rapidly as the noisy level increases. This is consistent with the constant gap between their ground truth accuracies (gap=10% between 100%, 90%, 80%). However, for the proposed discrepancy ratio, the gap between these three models decreases rapidly when the noise level increases from 0 to 0.4. This is an undesirable behavior since it over-estimate the performance gap in the low-noise regime. Since most real-world datasets could belong to this low-noise regime, this rapid change of performance gaps seems problematic.

In section 4.1.4, the claim that "It is in this regime that the model performance can be said to be better than the average human performance" seems questionable. When the ratio is less than 1, it simply means the annotator-model agreement is larger than the annotator-annotator agreement. What's the definition of the "average human performance" in this context?

In the work, multiple assumptions were made regarding different annotators. First, multiple annotators are assumed to be conditionally independent given the ground-truth label. Second, each annotator should provide high-quality labels. Third, different experts should have similar expertise. However, in medical applications, any of these assumptions could be violated. For example,1) multiple experts could provide similar labels given their similar knowledge background. It's more likely to observe multiple groups of experts, where experts in the same group are similar. 2), experts could provide random labels when they are extremely uncertain 3) different experts could have varying levels of expertise. Could the author provide some ideas on how to relax some of these assumptions?

Comments after reading authors' reply
-----------------------------------------------------
I would like to thank the authors for their detailed reply to my questions. The key novelty of this work, i.e., evaluating model performance (in terms of comparing different models and model vs average annotator performance) in the presence of noisy labels, should be appreciated. I'll raise my score to 6: Weak Accept. In the future work, it would be interesting to apply the proposed metric (or develop new metrics) to study whether any of the assumptions (conditional independence, reasonably high accuracy of each individual expert, different expert accuracies) can be relaxed.

**Experience Assessment:**

I have published one or two papers in this area.

**Review Assessment: Checking Correctness Of Derivations And Theory:**

I assessed the sensibility of the derivations and theory.

**Review Assessment: Checking Correctness Of Experiments:**

I carefully checked the experiments.

**Review Assessment: Thoroughness In Paper Reading:**

I read the paper thoroughly.

---

> ### Author Response · Authors · 2019-11-11
> **Response to Official Blind Review #1 (1/2)**
>
> We thank the reviewer for their helpful comments and suggestions and for bringing up several important issues. We have addressed in detail all of the reviewer’s comments below.
>
> “In the experimental results on MNIST classification, for Naive Evaluation and Majority Vote, as the author pointed out, the gap between three models (model 1.0, model 0.9, model 0.8) becomes narrower when the label swap probability increases, indicating it's becoming more difficult to differentiate model improvements. However, this is also true for the proposed discrepancy ratio, as is shown in Figure 2(b). Therefore, the discrepancy ratio didn't overcome those "drawbacks" associated with these two baselines, correct? In other words, in the high-noise regime, it would be natural to observe narrower gaps between those models.“
>
> We would like to clarify an important point that may have been unclear in our initial submission.The reviewer is referring to a comment made in Section 4.1.1 that “...as the label noise increases, the observed accuracy converges and it becomes increasingly difficult to differentiate model improvements”. This remark was meant to support the use of the Majority Vote method over the Naive Evaluation method, since majority vote can to some extent improve the ability to distinguish models by averaging away the label noise. We have clarified this point explicitly in the revised version of the paper.
>
> To be sure, we never claimed that the discrepancy ratio itself removes the narrowing gap between models as the label noise increases, and this was not our goal. By definition the performance gap between models has to narrow and converge to zero for *any* metric since when the swap probability is 1 the labels are completely random and hence all models are equivalent. If one simply wants to rank models (determine which one is better) than *any* of the baseline methods are perfectly suitable. On the other hand, as discussed in Section 2, in many real-world applications one instead needs to know how a model performs at a task relative to how humans perform. To answer this question, the Naive Evaluation and Majority Vote methods are no longer useful, whereas the discrepancy ratio is quite insightful.
>
> “In MNIST classification, the performances gap of those three models evaluated by Naive Evaluation and Majority Vote didn't decrease rapidly as the noisy level increases. This is consistent with the constant gap between their ground truth accuracies (gap=10% between 100%, 90%, 80%). However, for the proposed discrepancy ratio, the gap between these three models decreases rapidly when the noise level increases from 0 to 0.4. This is an undesirable behavior since it over-estimate the performance gap in the low-noise regime. Since most real-world datasets could belong to this low-noise regime, this rapid change of performance gaps seems problematic.”
>
> The reviewer is quite correct that the performance gaps between the majority vote and discrepancy ratio metrics are different. Ultimately, our goal is a metric that captures how the model performs relative to annotator disagreement. To this end, any gap in quantitative performance between two models should capture the degree to which the model predictions are closer or further to annotators than annotators are to one another. To clarify this, Figures 1b and 2b demonstrate a subtle but crucial feature of metrics sensitive to annotator disagreement: models exhibiting similar accuracies (even within 10%) can actually perform *very* differently when compared to human variability. In this sense, the discrepancy ratio is *correctly* picking up on real performance differences that the baseline metrics are simply not sensitive to. As a case-in-point, consider the example of Models 1.0 and 0.9 when the label swap probability is equal to 0.05. When looking at the baseline metrics (Native Evaluation and Majority Vote) the models only appear to be different by 10% (Figures 1a and 1b). However, when taking into account the annotator disagreement (Figure 2b) a completely different story emerges and one immediately sees that the models are in fact dramatically different: despite only a slight difference in accuracy, model 0.9 is twice as far from annotators as they are from each other, while model 1.0 is twice as close, thus resulting in a performance gap of 400%. We have clarified this very important point in Section 4.1.4.

---

> > ### Author Response · Authors · 2019-11-11
> > **Response to Official Blind Review #1 (2/2)**
> >
> > “In section 4.1.4, the claim that "It is in this regime that the model performance can be said to be better than the average human performance" seems questionable. When the ratio is less than 1, it simply means the annotator-model agreement is larger than the annotator-annotator agreement. What's the definition of the "average human performance" in this context?”
> >
> > Without access to unambiguous ground-truth labels, a natural way to define human performance is by comparing an annotator’s labels to those provided by other humans on the same samples. Consequently, we use the annotator discrepancy (Eq. 2) as a measure of average human performance. We have stated this explicitly in the revised paper immediately following Eq. 2 so that it is absolutely clear what we mean by this term.
> >
> > Despite not having access to unambiguous ground-truth in real-world problems, we have provided empirical support for the aforementioned choice using MNIST. The MNIST experiments demonstrate that the point at which the discrepancy ratio falls below 1 corresponds precisely to the point at which the model performance exceeds that of the (simulated) humans. The red and blue dots in Figure 2b occur at label swap probabilities of 0.1 * 10/9 and 0.2 * 10/9, respectively, which correspond precisely to when the accuracies of the simulated human annotators equal that of the models (0.9 and 0.8, respectively).
> >
> > “In the work, multiple assumptions were made regarding different annotators. First, multiple annotators are assumed to be conditionally independent given the ground-truth label. Second, each annotator should provide high-quality labels. Third, different experts should have similar expertise. However, in medical applications, any of these assumptions could be violated. For example,1) multiple experts could provide similar labels given their similar knowledge background. It's more likely to observe multiple groups of experts, where experts in the same group are similar. 2), experts could provide random labels when they are extremely uncertain 3) different experts could have varying levels of expertise. Could the author provide some ideas on how to relax some of these assumptions?”
> >
> > We completely agree with the reviewer regarding all three of these points! Thanks for helping us clarify the paper on these issues:
> >
> > 1). The reviewer is absolutely correct that there can be situations where annotators adhere to several distinct “schools of thought” and their annotations could be similar within each group but different between groups. In these cases it may make more sense to evaluate the discrepancy ratio independently with each group rather than lumping everyone together. We have added this point to the Limitations section of our paper (Section 5).
> >
> > 2). As we mentioned in Section 4.1.3, in many tasks annotators must be allowed to voluntarily skip examples they feel uncertain about precisely so that they don’t provide random labels. It is this skipping that can lead to the strict annotator bias that is resolved through the use of the discrepancy ratio (see Figure 2d). We have added a remark to the Limitations section that emphasizes that in difficult tasks annotators must be allowed to skip examples where they are unsure about the label.
> >
> > 3). We explicitly state in Limitations section that our approach assumes that the experts have been vetted as capable for this task. We have provided details of our vetting procedure in Appendix A and have added a reference to a comprehensive review paper on gathering reliable labels (Hovy and Lavid, Int. Journal of Translation (2010)) to the Limitations section.

---

### Official Review · AnonReviewer2 · 2019-10-21
**Official Blind Review #2**

**Rating:** 6

**Review:**

The authors presented an interesting paper which tries to solve a practically important question. Biomedical and tech industries usually hire human labelers for machine learning tasks, whose labels are usually noisy. Therefore, it is important to have a metric that can distinguish the performance of models with noisy labels.

In this paper, the authors proposed to measure the model performance based on the ratio of the discrepancy between the model prediction and the labeler, with the discrepancy between the labelers. The authors showed that in binary classification settings, their proposed metric can be reduced to simple to analyze quantities. The authors demonstrated the performance of their proposal in synthetic data as well in two real-world medical image datasets.

Overall, the paper is well motivated with a reasonable amount of novelty. The numerical experiments are well conducted, but I am not totally convinced by their results.

Detailed comments:
Significance: The paper is trying to address a practically important issue in machine learning.
Novelty: The mathematical formulation of the metric consists of simple sums and averages, which in itself is not novel. However, the authors' choice of using such formulations to assess model performance is novel.
Clarity: The authors could simplify their notations, and could periodically remind the readers about the notations. For example, I was stuck by the sigma^2 notation in (6) (undefined) and the m notation in (7) defined in the past page in small texts. It would help me a lot if the authors reminded me the definition of those notations when they appeared.
Numerical Experiments: This is my biggest complain. I wasn't able to conclude that the authors proposal is better than majority voting based on Figures 1b and 2b. To me they look qualitatively the same. Is there any reason that the discrepancy ratio is superior to the majority vote? In addition, I didn't see whether the curves in Figures 1 and 2 are an average of numerous simulated samples. if not, the authors should use the average to mitigate randomness; and if the curves are averages, then the distribution of the metrics should be described, because I think ultimately, we don't care about whether the average black curve is above the average red curve, but the chance of black curve being above the red curve.

**Experience Assessment:**

I have read many papers in this area.

**Review Assessment: Checking Correctness Of Derivations And Theory:**

I assessed the sensibility of the derivations and theory.

**Review Assessment: Checking Correctness Of Experiments:**

I assessed the sensibility of the experiments.

**Review Assessment: Thoroughness In Paper Reading:**

I read the paper at least twice and used my best judgement in assessing the paper.

---

> ### Author Response · Authors · 2019-11-11
> **Response to Official Blind Review #2**
>
> We thank the reviewer for their helpful comments and suggestions. We have addressed in detail all of the reviewer’s concerns and hope the resulting paper (uploaded) merits unambiguous acceptance. Our point-by-point response is detailed below.
>
> “Novelty: The mathematical formulation of the metric consists of simple sums and averages, which in itself is not novel. However, the authors' choice of using such formulations to assess model performance is novel.”
>
> The derivation of the model and annotator discrepancies in Section 3.3 was simply meant to serve as a concrete example of how these metrics are calculated in a particularly simple case. We hope that this example clarifies to the reader the more general formalism presented in Section 3.1 and helps to build intuition for the properties of the discrepancy ratio (see for example the last paragraph of Section 3.3). We would also like to point out that the novelty of our paper is not in this simple example but in providing a robust, task-independent and highly-interpretable metric that allows us to quantify model performance in the context of how humans perform relative to each other.
>
>
> “Clarity: The authors could simplify their notations, and could periodically remind the readers about the notations. For example, I was stuck by the sigma^2 notation in (6) (undefined) and the m notation in (7) defined in the past page in small texts. It would help me a lot if the authors reminded me the definition of those notations when they appeared.”
>
> Thanks for pointing this out. We have added to the revised manuscript the definition of the variance to Eq. 5 and included several additional reminders of the meaning of symbols when they occur far from their definitions.
>
>
> “Numerical Experiments: This is my biggest complain. I wasn't able to conclude that the authors proposal is better than majority voting based on Figures 1b and 2b. To me they look qualitatively the same. Is there any reason that the discrepancy ratio is superior to the majority vote?
>
> This is a subtle point that could have been made more clear in our initial submission. The discrepancy ratio is indeed superior to the majority vote method, as discussed in detail in Section 2, Section 4.1.2 and Section 4.2.1, though this may not be entirely clear from simply looking at Figures 1b and 2b. The curves in these figures look qualitatively similar only in the sense that both allow us to decide which model is better. Indeed, if the goal is simply to rank models, then *any* of the baseline metrics are perfectly suitable. If, however, one needs to know how a particular model performs *relative to humans*, the majority vote method simply does not answer that question. This is the main motivation for using the discrepancy ratio and has been clarified in Section 4.1.2.
>
> In other words, evaluating against majority vote does not give a clear indication of whether a machine learning model is reasonably substitutable for humans who are currently performing a given task in the real world. This is especially important in medical applications since decisions in the field are usually made by a single individual, not majority consensus. The discrepancy ratio, on the other hand, unambiguously answers the question of how a model performs against the humans by considering human-to-human disagreement. An additional property that makes the discrepancy ratio superior to the majority vote method is that the majority vote method gives a metric that is dependent on the number of annotators that annotated the dataset (as can be inferred from Figures 1a and 1b), while the discrepancy ratio is independent of the number of annotators (Figure 2c).
>
>
> “In addition, I didn't see whether the curves in Figures 1 and 2 are an average of numerous simulated samples. if not, the authors should use the average to mitigate randomness; and if the curves are averages, then the distribution of the metrics should be described, because I think ultimately, we don't care about whether the average black curve is above the average red curve, but the chance of black curve being above the red curve.“
>
> The curves in Figures 1 and 2 are generated by taking the MNIST test set and randomly swapping the labels with probability p, as described in Section 4.1. Each point on these plots is generated with an independent swapping of the labels resulting in the small fluctuations that are evident in these curves. As requested by the reviewer, in the revised version of the paper we have replaced the curves with their averages as well as overlaid 95% quantiles to indicate the distributions of all the metrics.

---

### Decision · Program_Chairs · 2019-12-19

**Decision:**

Accept (Poster)

**Comment:**

This paper tackles an interesting problem: "How should we evaluate models when the test data contains noisy labels?". This is a particularly relevant question in the medical imaging domain where expert annotators often disagree with each other. The paper proposes a new metric "discrepancy ratio" which computes the ratio how often the model disagrees with humans to how often humans disagree with each other. The paper shows that under certain noise models for the human annotations the discrepancy ratio can exactly determine when a model is more accurate than humans, whereas commonly used baselines such as comparing with the majority vote do not have this property. Reviewers were satisfied with the author rebuttal, particularly with the clarification that the goal of the metric is to accurately determine when model performance exceeds that of human annotators, and not to better rank models. The metric should be quite useful, assuming users are cautious of the limitations described by the authors.